# A cancer vaccine-mediated postoperative immunotherapy for recurrent and metastatic tumors

Tingting Wang[1,2], Dangge Wang[1,2], Haijun Yu [1], Bing Feng[1,2], Fangyuan Zhou[1], Hanwu Zhang[1], Lei Zhou[1], Shi Jiao[3] & Yaping Li[1]

Vaccines to induce effective and sustained antitumor immunity have great potential for postoperative cancer therapy. However, a robust cancer vaccine simultaneously eliciting tumor-specific immunity and abolishing immune resistance continues to be a challenge. Here we present a personalized cancer vaccine (PVAX) for postsurgical immunotherapy. PVAX is developed by encapsulating JQ1 (a BRD4 inhibitor) and indocyanine green (ICG) co-loaded tumor cells with a hydrogel matrix. Activation of PVAX by 808 nm NIR laser irradiation significantly inhibits the tumor relapse by promoting the maturation of dendritic cells and eliciting tumor infiltration of cytotoxic T lymphocytes. A mechanical study reveals that NIR light-triggered antigen release and JQ1-mediated PD-L1 checkpoint blockade cumulatively contribute to the satisfied therapeutic effect. Furthermore, PVAX prepared from the autologous tumor cells induces patient-specific memory immune response to prevent tumor recurrence and metastasis. The PVAX model might provide novel insights for postoperative immunotherapy.

[1] State Key Laboratory of Drug Research & Center of Pharmaceutics, Shanghai Institute of Materia Medica, Chinese Academy of Sciences, 201203 Shanghai, China. [2] University of Chinese Academy of Sciences, 100049 Beijing, China. [3] Institute of Biochemistry and Cell Biology, Shanghai Institutes for Biological Sciences, Chinese Academy of Sciences, 200031 Shanghai, China. These authors contributed equally: Tingting Wang and Dangge Wang. Correspondence and requests for materials should be addressed to H.Y. (email: hjyu@simm.ac.cn) or to Y.L. (email: ypli@simm.ac.cn)

Surgical resection is the primary option for clinical treatment of early stage and nonmetastatic solid tumors. However, residual microtumors or circulating tumor cells (CTCs) may cause lethal tumor recurrence and metastasis for months or years postoperation[1, 2]. Neoadjuvant chemotherapy and radiotherapy are the current standard-of-care for post-surgical cancer management, they both tend to decrease the life quality of patients[3]. Alternatively, approaches utilizing cancer immunotherapy have emerged for tumor regression and metastasis prevention by stimulating the host immune response[4–10]. Immunotherapy using peptide-based vaccines or checkpoint inhibitors has induced durable clinical response in several types of tumors including melanoma, lung and bladder cancers[11, 12]. The recent innovation in cancer exome sequencing has further promoted the development of patient-specific neoantigens for personalized immunotherapy[13–17]. Unfortunately, the objective response rates of peptide-based vaccines have been disappointing in clinical trials due to tumor heterogeneity and their inefficiency to co-deliver multiple antigen peptides and adjuvants to the draining lymph nodes (LNs)[7, 18, 19]. Furthermore, the sequencing and preparation of patient-derived neoantigens is costly and time consuming, which is unfavorable for metastatic or late-stage cancer patients[20].

Cancer vaccines developed from whole tumor cells represent a promising avenue for cancer immunotherapy. Identification of the tumor antigens is not necessary for tumor cell vaccine generation since the tumor cells contain the whole array of mutated epitopes for parallel presentation to both CD8[+] and CD4[+] T cells, which reduces the likelihood of tumor escape[21]. Moreover, the autologous tumor cell source obtained from patients can release patient-specific antigens to trigger antitumor immunity and achieve personalized immunotherapy[22, 23]. However, the tumor cell vaccine-initiated immune response is negatively modulated in the intrinsic immunosuppressive tumor microenvironment[7, 15, 24–26]. Moreover, tumor cell vaccines can induce interferon γ (IFN-γ) secretion, which elicits the expression of programmed death ligand 1 (PD-L1) in the tumor and thus induces adaptive immune resistance[27].

To address the above issues associated with vaccine-based cancer immunotherapy approaches, we herein propose a personalized cancer vaccine (PVAX) with immune checkpoint blockade capacity for postsurgical immunotherapy of the recurrent and metastatic tumors. PVAX is developed by integrating JQ1 and indocyanine green (ICG) co-loaded tumor cells with a hydrogel matrix (Fig. 1a). JQ1 is a small molecular inhibitor for bromodomain and extraterminal (BET) protein BRD4, which overcomes immune tolerance by suppressing intratumoral expression of PD-L1[25, 28]. ICG is a well-studied photoabsorbent with high photothermal conversion efficiency, which has been widely exploited for photothermal therapy of various solid tumors[7, 29, 30]. Upon NIR laser irradiation, ICG induces significant temperature elevation to trigger on-demand release of tumor-specific antigens and JQ1. The hydrogel matrix is fabricated with a tumor-penetrable peptide to prevent cargo leakage and facilitate tumor penetration of the tumor vaccines for overcoming tumor burden (Fig. 1b). We demonstrate that administration of the PVAX through local injection followed by NIR laser-triggered activation efficiently prevent postoperative tumor recurrence and metastasis by simultaneously boosting patient-specific immune responses and blocking PD-L1-dependent immune evasion. The simple PVAX fabrication procedure combined with its ability to simultaneously induce a patient-specific immune response, and combat immunological resistance, supports its potential as a robust cancer vaccine for post-surgical immunotherapy.

## Results

**Fabrication of PVAX.** To fabricate the PVAX for personalized immunotherapy, tumor cells from mouse 4T1 breast tumor xenografts were collected and fixed in a Foxp3 cell fixation and permeabilization buffer. The fixed tumor cells were then incubated in a methanol solution of ICG and JQ1 for 4 h to obtain

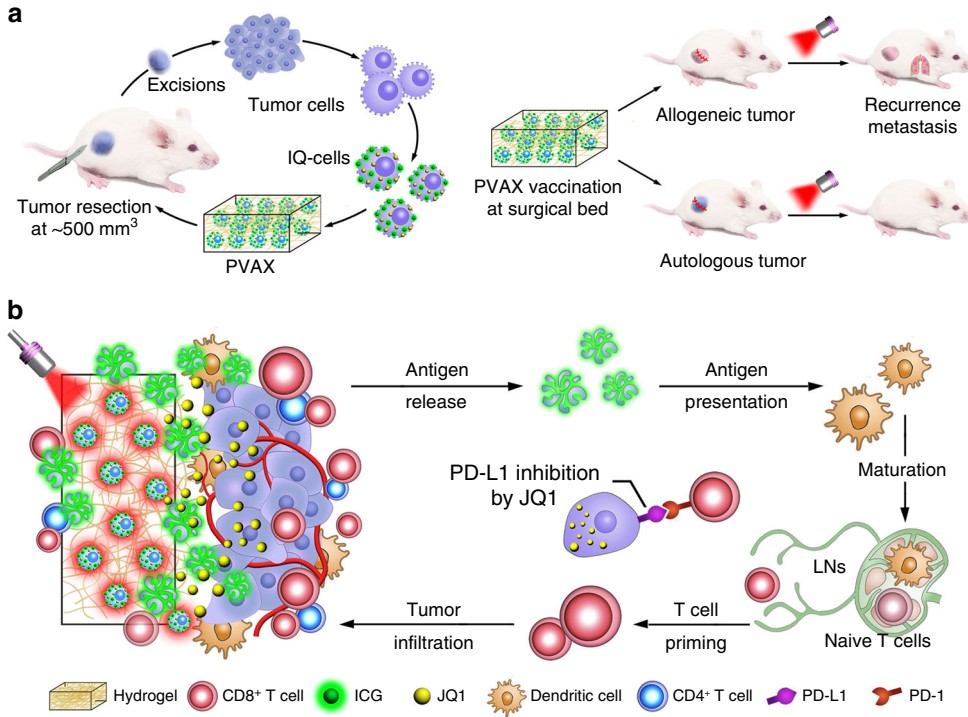

**Fig. 1** Schematic illustration of fabrication of PVAX for cancer immunotherapy. **a** Fabrication process of PVAX. **b** Simplified mechanism of PVAX-mediated cancer immunotherapy to prevent post-operative tumor recurrence and metastasis

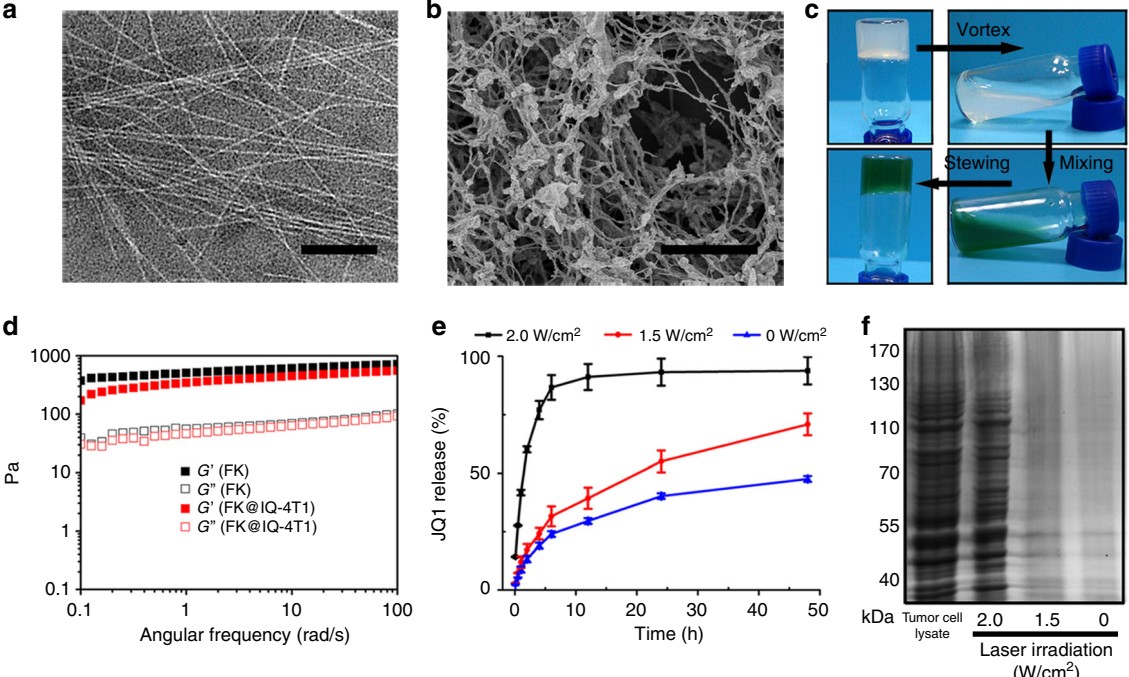

**Fig. 2** Characterization of PVAX for cancer immunotherapy. **a** TEM (Scale bar, 100 nm), and **b** SEM (Scale bar, 20 μm) examination of the tumor cell-loaded FK hydrogel. **c** Hydrogel encapsulation of ICG and JQ1 co-loaded 4T1 tumor cells (IQ-4T1). **d** Frequency-dependent rheological properties of pure and FK@IQ-4T1 tumor hydrogels examined at 37 °C (peptide concentration 10 wt%, strain 1%). **e** NIR laser-triggered JQ1 release profile of FK@IQ-4T1 as a function of photodensity. Data represent means ± s.d. (*n* = 3). **f** SDS-PAGE electrophoresis of laser-triggered tumor antigen release from FK@IQ-4T1vaccine, the lysate of whole 4T1 cells was used as a positive control. FK@IQ-4T1 vaccine were illuminated with 808 nm laser for 2 min at the desired photodensity

ICG and JQ1 co-loaded tumor cells (abbr., IQ-4T1). The ICG and JQ1 loading ratios were determined to be ~18.6 ± 0.4 and 26.3 ± 0.7 μg/10⁶ tumor cells using UV–Vis photospectrometry and high-performance liquid chromatography (HPLC) measurements, respectively (datas represent means ± s.d. (*n* = 3), Supplementary Fig. 1). The IQ-4T1 cells were further encapsulated with a hydrogel matrix to prevent payload leakage. The hydrogel was prepared with a tumor-penetrable peptide sequence of Fmoc-KCRGDK (FK). FK was employed to facilitate tumor penetration of the tumor antigens and JQ1 by binding with neuropilin-1 (Nrp-1), a protein overexpressed on surface of the tumor cells[31–33]. Two Fmoc groups were grafted on the peptide sequence to stabilize the peptide hydrogel via π–π stack and hydrophobic interaction among the Fmoc groups. The Fmoc groups were modified on the N-terminal of the peptide to avoid impairing the tumor penetration activity of the CRGDK motif. Fmoc-KCRGDC (FC) peptide was also synthesized as an tumor non-penetrable analog of FK for vaccine development (Supplementary Fig. 2, 3).

FK peptide formed self-assembled micelles in PBS, which transformed to assembled nanofibers, and to hydrogels after 70 °C incubation for 8.0 min as examined using transmission electron microscopy (TEM) (Fig. 2a and Supplementary Fig. 4a), scanning electron microscopy (SEM) (Fig. 2b and Supplementary Fig. 4b) and optical transmittance measurements (Supplementary Fig. 4c–f). The resultant FK hydrogel was thixotropic and injectable due to the noncovalent crosslinking nature of the self-assembling materials, allowing IQ-4T1 encapsulation through simple mixing, followed by local injection of the cancer vaccines at the surgical bed (Fig. 2c). The IQ-4T1-loaded FK hydrogels (termed as FK@IQ-4T1) displayed similar rheological properties as cell-free hydrogels, indicating encapsulation of tumor cells negligibly affected the mechanical

properties of the peptide hydrogels (Fig. 2d and Supplementary Fig. 4g).

To determine the distribution of the tumor cells inside the hydrogel matrix, JQ1 was replaced with the fluorescent dye Coumarin 6 (C6) to obtain ICG and C6 co-loaded FK@IC-4T1 hydrogels. The fluorescence distribution of C6 and ICG was then examined by using confocal laser scanning microscopy (CLSM). The CLSM images revealed that the green fluorescent dots of C6 and the red fluorescent dots of ICG distributed evenly within the hydrogel matrix and colocalized well with the cellular nucleus (blue, stained with DAPI, Supplementary Fig. 5a). Upon laser irradiation with 808 nm NIR light, FK@IQ-4T1 induced a noticeable hyperthermia effect as a function of ICG concentration or photodensity (Supplementary Fig. 5b & c). Laser irradiation dramatically triggered the release of JQ1 and tumor antigens via hyperthermia-induced hydrogel dissociation and thermal lysis of the cell membrane (Fig. 2e, f)[34]. JQ1 released from PVAX maintained its structural integrity as confirmed by high performance liquid chromatography (HPLC) and mass spectroscopy (MS) (Supplementary Fig. 6).

**NIR light promotes the release of JQ1 and tumor-specific antigens in vitro.** To examine NIR light-triggered JQ1 and antigen release from the cancer vaccines in vitro, NIR light-promoted intratumoral penetration of FK@IC-4T1 was first examined in 4T1 tumor cell-derived multicellular spheroids (MCSs). FC@IC-4T1 without tumor penetration ability was prepared for negative control. The C6 fluorescence of the laser-treated IC-4T1 and FC@IC-4T1 groups was restricted to the outer layer of MCSs, probably due to the elevated interstitial fluid pressure in the tumor spheroid (Supplementary Fig. 7). In contrast, the C6 fluorescence in the laser-treated FK@IC-4T1 group

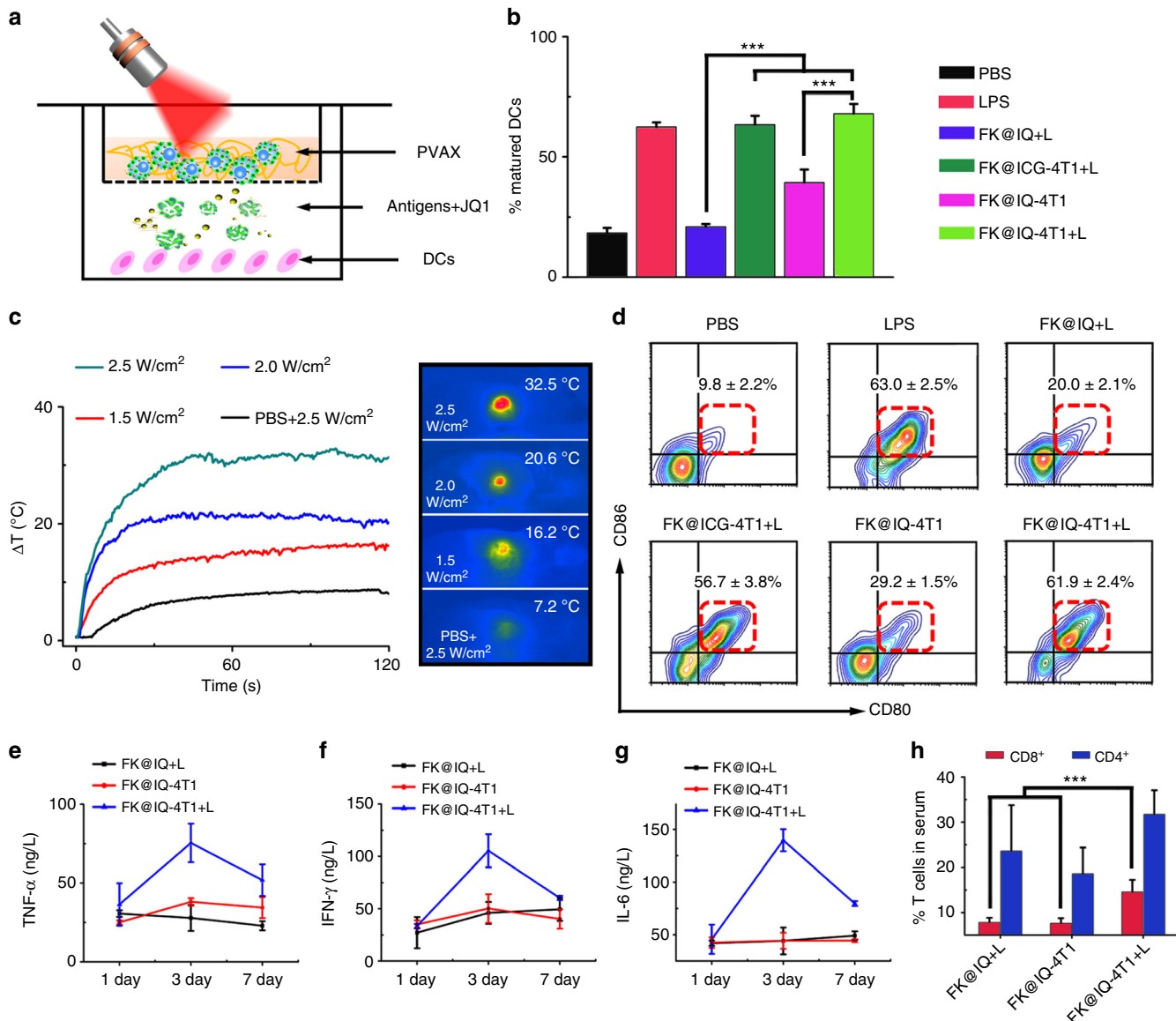

**Fig. 3** PVAX promoted DC maturation and induced immune response in vitro and in vivo. **a** Transwell co-culture set-up for inducing dendritic cell (DC) maturation in vitro. **b** PVAX-induced DC maturation (CD11c[+]CD80[+]CD86[+]) in vitro, the vaccines was illuminated with 808 nm laser for 2.0 min at a photodensity of 2.0 W/cm[2]. Unpaired student's t-test (two-tailed) was used for comparison between two groups. ***p < 0.01. Data represent mean ± s.d. (n = 3). **c** Photothermal effect of FK@IQ-4T1 vaccine in vivo. **d** The frequency of mature DCs (CD11c[+]CD80[+]CD86[+]) in draining LNs of BALB/c mice upon different treatments. Data represent mean ± s.d. (n = 3). **e, g** Serum concentrations of **e** TNF-α, **f** IFN-γ, and **g** IL-6 examined at the desired time points post treatment. **h** The frequency of serum CD8[+] and CD4[+] T cells examined 3 days post the indicated treatments. Unpaired student's t-test (two-tailed) was used for comparison between two groups. ***p < 0.01. Data represent mean ± s.d. (n = 3)

was dramatically increased and diffused throughout the MCSs, demonstrating intratumoral penetration of C6 payloads activated by laser-triggered FK release and FK-facilitated intratumoral penetration of C6 (Supplementary Fig. 8). The ability of FK-based PVAX to efficiently penetrate tumors is a significant improvement over that of conventional cancer vaccines since their therapeutic efficacy in large tumors was severely impaired by their inefficiency to combat the tumor burden[35–37].

The in vivo tumor penetration property of PVAX was next investigated in a 4T1 tumor-bearing BALB/c mouse model. FK@IC-4T1, FC@IC-4T1 or IC-4T1 was subcutaneously (s.c.) injected into the 4T1 tumor xenograft when the tumor volume reached 100 mm[3]. Fluorescence imaging in vivo 120 h post-injection revealed that the FC@IC-4T1 and FK@IC-4T1 groups were of 2.8-fold and 2.4-fold higher ICG fluorescence intensity

than that of the IC-4T1 group respectively, suggesting hydrogel encapsulation remarkably enhanced the tumor accumulation and retention of ICG (Supplementary Fig. 9). Upon 808 nm laser irradiation, the C6 fluorescence of the FC@IC-4T1 vaccine was restricted mainly to the peritumoral areas. In contrast, C6 was released from the FK@IC-4T1 vaccine and diffused throughout the tumor sections (Supplementary Fig. 10). The fluorescence imaging data in vitro and ex-vivo confirmed that FK effeiciently overcame the tumor burden and facilitated the tumor penetration of C6 by the binding of the CRGDK motif with the membrane Nrp-1 of the tumor cells and triggering an active transport process[33, 38]. The FK peptide also showed good biocompatibility in RAW 264.7 macrophages and bone marrow derived dendritic cells (BMDCs) (Supplementary Fig. 11), implying its good potential for in vivo application.

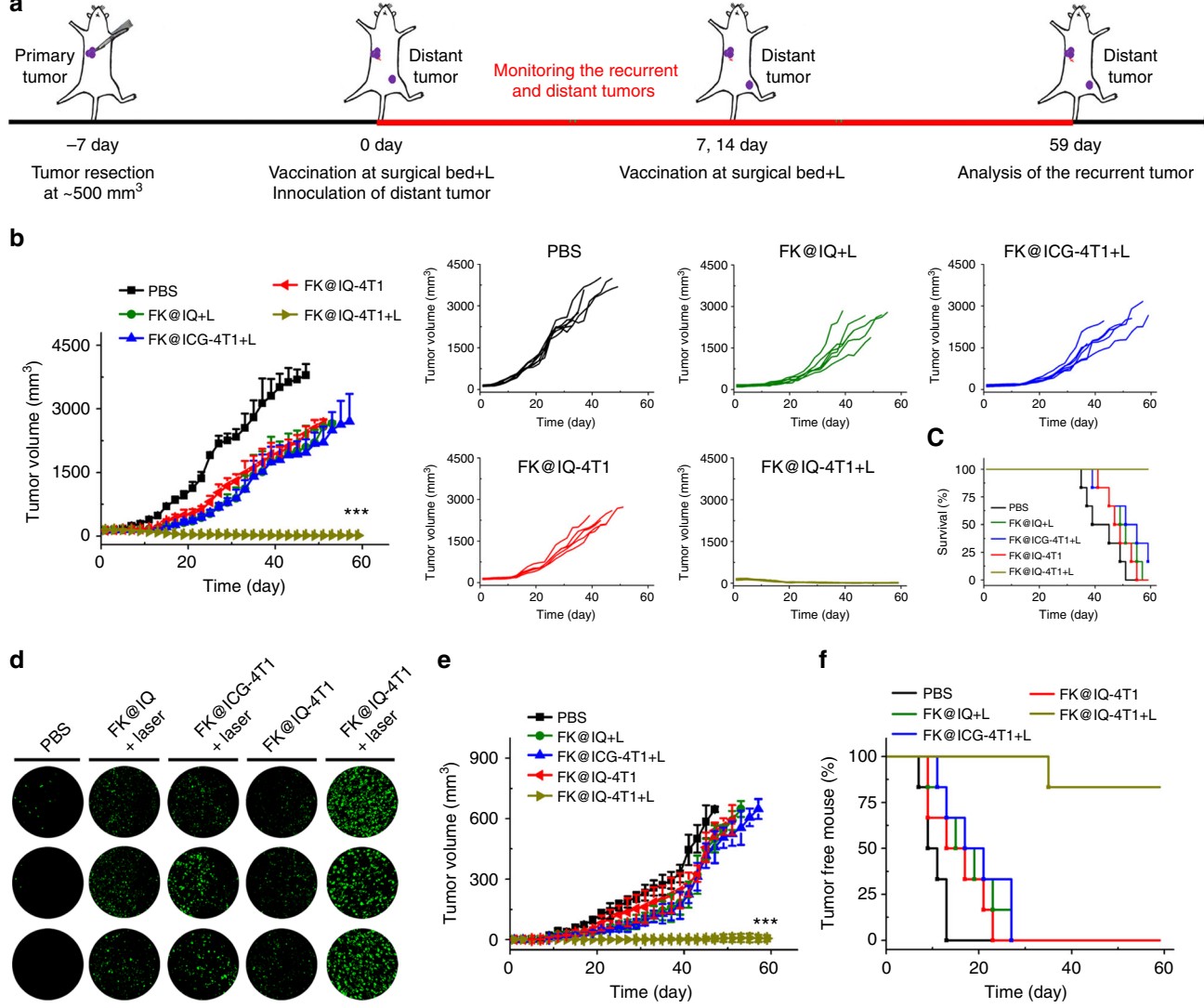

**Fig. 4** PVAX-performed immunotherapy in recurrent and metastatic 4T1 xenografts. **a** Treatment schedule for the immunotherapy. **b** Average and individual tumor growth curves of the recurrent tumors receiving different treatments (laser irradiation for 2 min at photodensity of 2.0 W/cm$^2$). The mean tumor volumes were analysed using one-way ANOVA. ***$p < 0.01$. Data represent mean ± s.d. ($n = 6$). **c** Survival percentage of the 4T1 recurrent tumor-bearing BALB/c mice receiving different treatments. **d** TUNEL staining of the tumor sections examined at the end of antitumor study. **e** Growth inhibition of the distant tumors via PVAX+L. The mean tumor volumes were analysed using one-way ANOVA. ***$p < 0.01$. Data represent mean ± s.d. ($n = 6$). **f** Tumor-free percentages of the BALB/c mice rechallenged with the distant tumor

**PVAX shows immune-stimulation abilities**. Dendritic cells (DCs) play crucial roles in initiating and regulating the innate and adaptive immune response[39]. The interaction between the FK@IQ-4T1 vaccine and the immune system was evaluated by examining DC maturation in vitro and in vivo. BMDCs separated from the BALB/c mice were incubated with FK@IQ-4T1 for 24 h and the frequency of matured DCs (CD11c$^+$CD80$^+$CD86$^+$) was then examined using flow cytometry (Fig. 3a, b). FK@IQ-4T1 alone induced moderate DC maturation after 24 h incubation. In contrast, laser irradiation significantly promoted the maturation of FK@IQ-4T1-incubated DCs, which could be attributed to laser-triggered release of tumor antigens. It is worth noting that FK@ICG-4T1+L without JQ1 loading promoted DC maturation as efficient as FK@IQ-4T1+L, suggesting that JQ1 itself does not elicit the immune response.

To investigate whether FK@IQ-4T1 vaccination accelerated DC maturation in vivo, mice were vaccinated with FK@IQ, FK@ICG-4T1 or FK@IQ-4T1. The injection sites were illuminated with 808 nm laser to induce the hyperthermia effect for

triggering JQ1 and tumor antigen release in a spatiotemporally controlled manner (Fig. 3c). The mice were then sacrificed 3 days post vaccination to harvest the draining LNs for flow cytometric analysis of DC maturation. Combined FK@IQ-4T1 vaccination and laser irradiation significantly promoted up to ~62% of DC maturation, which was 3.1-fold and 2.1-fold more efficient than 4T1 cell-free FK@IQ+L and FK@IQ-4T1, respectively (Fig. 3d).

The DC maturation-induced systemic immune response was further evaluated by measuring the serum concentration of cytokines and pro-inflammatory mediators using enzyme-linked immunosorbent assays (ELISA). Compared to FK@IQ and FK@IQ-4T1 controls, 808 nm laser illumination sustainably promoted the secretion of IFN-γ, tumor necrosis factor α (TNF-α), and interleukin 6 (IL$^-$6) in the FK@IQ-4T1 group (Fig. 3e–g). For instance, the serum levels of TNF-α, IFN-γ and IL-6 in the FK@IQ-4T1+L group were 2.7-fold, 2.3-fold, and 3.1-fold higher than those of the FK@IQ-4T1 group examined 3 days post second vaccination. Additionally, upon FK@IQ-4T1+L treatment, the serum frequency of CD8$^+$ cytotoxic T lymphocytes

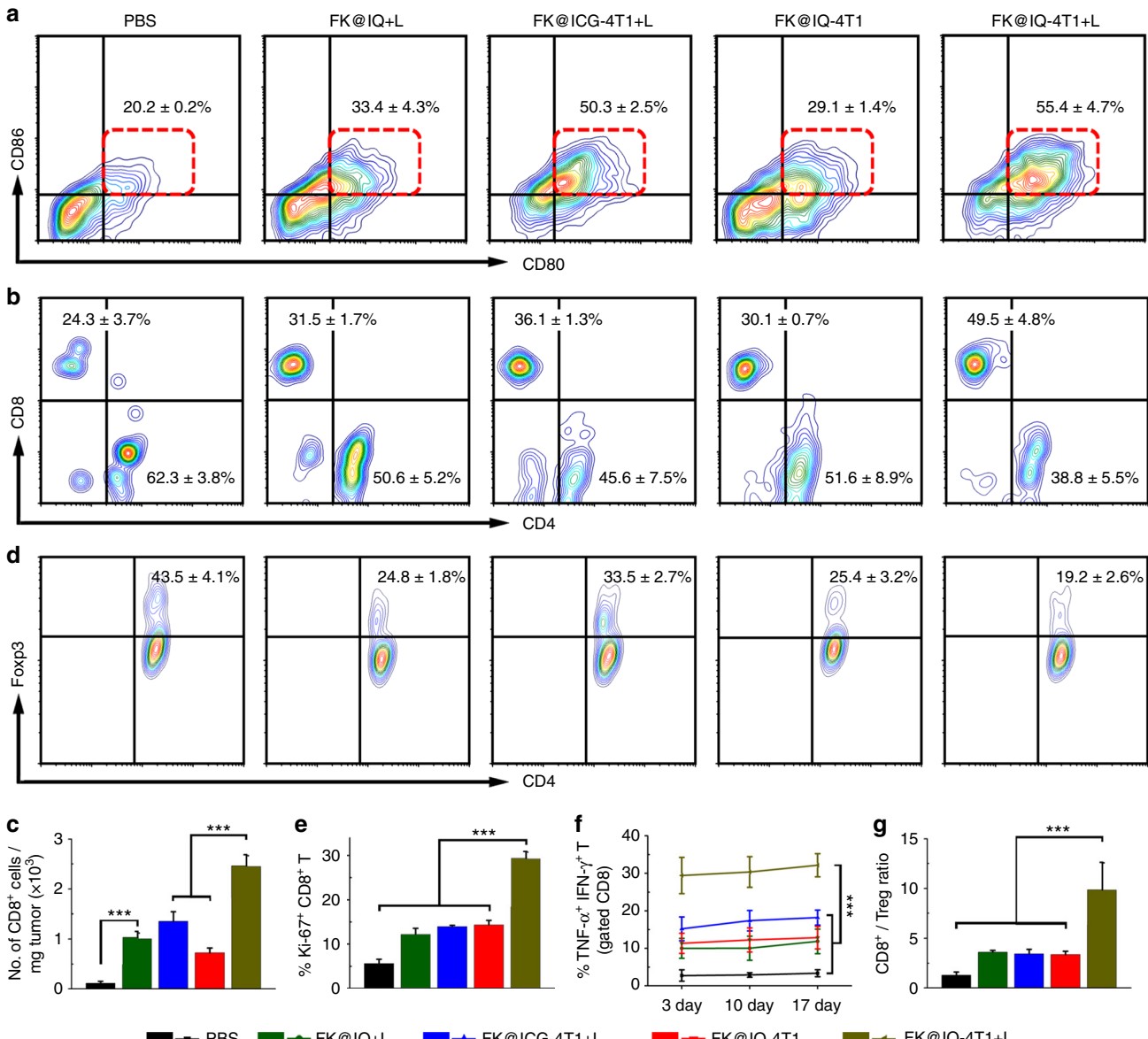

**Fig. 5** Vaccine-induced antitumor immunity in vivo. **a** The hydrogel vaccines accelerated DC maturation (CD11c$^+$CD80$^+$CD86$^+$) in the tumor examined 3 days post treatment. Data represent mean ± s.d. ($n = 3$). **b** Flow cytometric examination of the intratumoral infiltration of CD4$^+$ and CD8$^+$ T cells (gated on CD3$^+$ T cells). Data represent mean ± s.d. ($n = 3$). **c** Tumor mass-normalized intratumoral infiltration of CD8+ T cells in the recurrent tumors examined 10 days post treatment. The comparison of two groups was followed by Unpaired student's $t$-test (two-tailed). ***$p < 0.01$. Data represent mean ± s.d. ($n = 3$). **d** The frequency of Tregs in the recurrent tumors after different treatments examined 10 days post treatment. Data represent mean ± s.d. ($n = 3$). **e** The proliferation activity of CD8$^+$ T cells in the recurrent tumor examined 10 days post treatment. **f** The frequency of TNF-α$^+$/IFN-γ$^+$ CD8$^+$ T cells in the recurrent tumors after different treatments examined 3 days post treatment. **g** Ratios of tumor-infiltrating CD8$^+$ T cells vs. Tregs in the recurrent tumor examined 10 day post the first treatment. **e**–**g** The significance of the deferences was evaluated by one-way ANOVA. ***$p < 0.01$. Data represent mean ± s. d. ($n = 3$)

(CTLs) significantly increased (Fig. 3h). These findings verified that laser irradiation efficiently promoted DC maturation and elicited the immune responses in vitro and in vivo by triggering antigen release from the FK@IQ-4T1 vaccine.

NIR light-mediated photothermal therapy using carbon nanotubes has been employed by Liu et al. for cancer immunotherapy by activating antitumor immunity[40]. We observed that FK@IQ+L induced a hyperthermia effect comparable to FK@IQ-4T1+L (Supplementary Fig. 12). However, the latter showed much higher efficiency to promote DC maturation, suggesting the immune response was caused by laser-triggered release of tumor antigens from FK@IQ-4T1, and not from the photothermal effect of ICG.

The intratumoral secretion of pro-inflammatory cytokines, particularly IFN-γ, has been shown to elicit PD-L1 expression in tumor-infiltrating lymphocytes (TILs) and tumor cells[25]. The upregulation of PD-L1 can in turn induce acquired immune tolerance by binding to programmed cell death receptor 1 (PD-1) expressed on the cell surface of CTLs[41]. Flow cytometric measurement revealed that a JQ1 concentration of 200 nM completely reversed IFN-γ-stimulated PD-L1 activation in 4T1 cells (Supplementary Fig. 13) by inhibiting BRD4 activation. Furthermore, JQ1 treatment efficiently suppressed PD-L1 expression in 4T1 tumor xenografts in vivo as examined by immunohistochemistry (Supplementary Fig. 14), providing direct

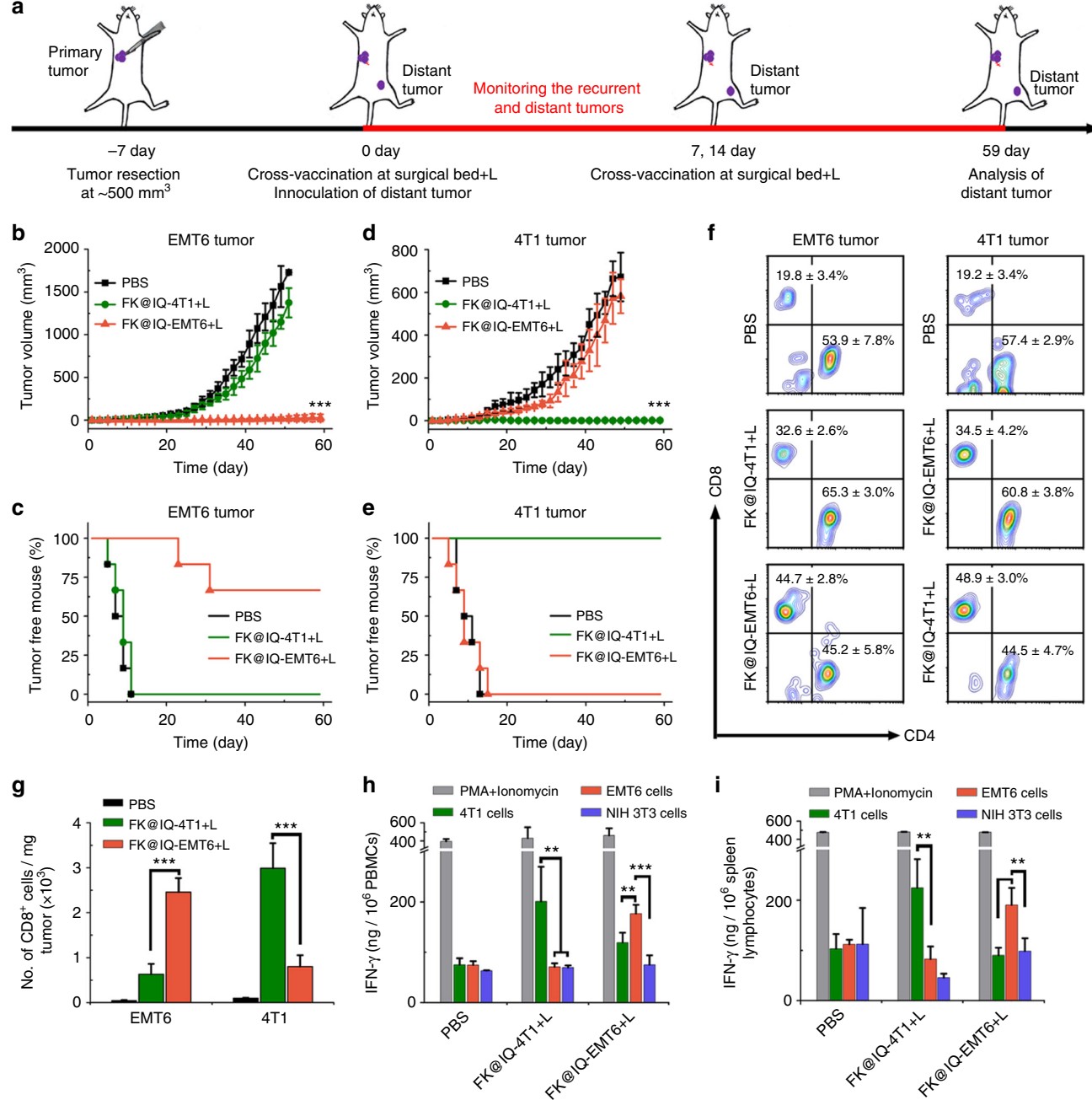

**Fig. 6** Cross-vaccination of 4T1 and EMT6 tumor xenografts by autologous or allogeneic cancer vaccines. **a** Cross-vaccination experimental design. **b**, **c** Distant tumor growth curves and tumor-free percentages of the distant EMT6 tumor-bearing BALB/c mice model. The mean tumor volumes were analysed using one-way ANOVA. ***$p < 0.01$. Data represent mean ± s.d. ($n = 6$). **d**, **e** Distant tumor growth curves and tumor-free percentages of the distant 4T1 tumor-bearing BALB/c mice. The mean tumor volumes were analysed using one-way ANOVA. ***$p < 0.01$. Data represent mean ± s.d. ($n = 6$). **f** Representative flow cytometry plots showing the proportions of tumor-infiltrating CTLs in distant EMT6 or 4T1 tumors. Data represent mean ± s.d. ($n = 3$). **g** Tumor mass-normalized intratumoral infiltration of CD8$^+$ T cells in distant EMT6 or 4T1 tumors. **h**, **i** Vaccination-induced secretion of tumor specific IFN-γ by the **h** peripheral blood mononuclear lymphocytes (PMBCs), and **i** the spleen lymphocytes, respectively. **g**–**i** Unpaired student's $t$-test (two-tailed) was used for the comparison between two groups. ***$p < 0.01$. Data represent mean ± s.d. ($n = 3$)

evidence for JQ1-mediated blockade of the PD-1/PD-L1 checkpoint in vivo.

### PVAX inhibits the postoperation tumor recurrence and metastasis.

The high efficacy of FK@IQ-4T1+L to promote DC maturation in vitro and in vivo prompted us to address whether the FK@IQ-4T1 vaccine induced antitumor immune response can prevent postsurgical tumor recurrence and metastasis. The orthotopic 4T1 tumor models were established by s.c. implanation of the 4T1 tumor cells into the right mammary fat pad of BALB/c mice. The majority of the primary tumor was surgically resected when the tumor volume reached 500 mm³. The surgical bed was then vaccinated with PBS, FK@IQ, FK@IQ-4T1 or FK@ICG-4T1 at an identical ICG dose of 2.5 mg/kg and JQ1 dose of 3.4 mg/kg when the recurrent tumors reached 100 mm³. Two hours post vaccination, the surgical bed of the FK@IQ, FK@ICG-4T1 and FK@IQ-4T1 groups was illuminated with an 808 nm

laser for 2 min. The treatments were repeated for three times at an time interval of six days. The ability of the PVAX to prevent tumor relapse was evaluated by monitoring the tumor growth every two days after the first treatment (Fig. 4a). FK@IQ-4T1 and FK@ICG-4T1+L delayed 36.4 and 48.1% of the tumor recurrence, respectively (Fig. 4b). In contrast, FK@IQ-4T1+L completely regressed the recurrent tumor in the whole experimental period of 59 days without mouse death and causing obvious body weight loss (Fig. 4c and Supplementary Fig. 15), indicating that laser-triggered JQ1 release from FK@IQ-4T1 significantly amplified the PVAX-induced antitumor immune response. Terminal deoxynucleotidyl transferase dUTP nick-end labeling (TUNEL) staining of the tumor sections showed that laser irradiation of the FK@IQ-4T1 group induced significant apoptosis and proliferation inhibition of the tumor cells (Fig. 4d and Supplementary Fig. 16), likely mechanisms to explain the anti-tumor effect of FK@IQ-4T1+L treatment. This phenomenon could be caused by the antitumor adaptive immune response since laser irradiation triggering antigen and JQ1 release promoted intratumoral infiltration and proliferation of effector CTLs.

To verify that FK@IQ-4T1+L could induce a systemic immune response to inhibit distant tumor metastasis, mice were challenged with $1 \times 10^5$ 4T1 tumor cells into the left flank of each mouse to establish a distant tumor after the first vaccination and light irradiation of the primary tumor post-surgery. The surgical beds of the primary tumors were continually treated by FK@IQ-4T1+L for two cycles at a time interval of 6 days, and the growth of the distant tumor was monitored during the whole therapeutic period. FK@ICG-4T1+L and FK@IQ+L inhibited 27.3% and 26.7% of the distant tumor growth, respectively (Fig. 4e). In contrast, FK@IQ-4T1+L highly efficiently eradicated the distant tumor grouth with a tumor free percentage of 83.3% (Fig. 4f), indicating that FK@IQ-4T1+L induced a robust antitumor immunity.

To demonstrate the crucial rule of JQ-1 for highly efficient immunotherapy, the therapeutic efficacy of the vaccine hydrogel was further investigated by s.c. vaccinaiton at the distant sites. The results showed that treatment by FK@IQ-4T1+L at the distant site moderately inhibited the recurrence of 4T1 primary tumor (Supplementary Fig. 17a–c), suggesting tumor-specific delivery of JQ-1 is critical for tumor recurrence prevention by suppressing PD-L1-mediated immune evasion. Interestingly, FK@IQ-4T1+L at the distant site completely eradicated the distant tumor, which could be attributed to treatment-induced systemic immune response (Supplementary Fig. 17d, e).

To better understand the mechanisms underlying the improved antitumor effect of the combination of FK@IQ-4T1 vaccination and NIR laser irradiation, the laser-promoted DC maturation in the tumor and intratumoral infiltration of CD8$^+$ CTLs were both examined by flow cytometry. Compared to FK@IQ-4T1, laser irradiation dramatically promoted the intratumoral DC maturation and infiltration of CD8$^+$ CTLs in the FK@IQ-4T1 vaccinated mice (Fig. 5a, b and Supplementary Fig. 18). The frequency of tumor infiltrating CD8$^+$ CTLs in the FK@IQ-4T1+L group was 21.8-fold higher that of the PBS control, and 3.4-fold higher than that of the FK@IQ-4T1 group (Fig. 5c). This confirmed that the combination of FK@IQ-4T1 vaccination and laser irradiation dramatically activated the antitumor immune response.

It was worth noting that although FK@ICG-4T1+L and FK@IQ-4T1+L showed comparable efficacy to induce DC maturation in the LNs and the tumor xenografts via laser-triggered release of tumor antigens (Figs. 3d, 5a). FK@IQ-4T1+L marginally increased the intratumoral infiltration of CD8$^+$ CTLs (Fig. 5b, c), further verifying the key role of JQ1 for PD-L1/PD-1 blockade and CTL activation in vivo.

We then investigated the intratumoral infiltration of regulatory T cells (Tregs, CD3$^+$CD4$^+$CD25$^+$Foxp3$^+$) was examined by flow cytometric examination since Tregs are capable of restraining effective anti-tumor immune responses of CTLs[42]. Figure 5d demonstrated that the Treg frequency in the vaccinated groups (e.g., FK@IQ-4T1 or FK@IQ-4T1+L) was much lower that of the PBS group, verifying FK@IQ-4T1+L significantly suppressed the immune tolerance of the recurrent tumor.

To further demonstrate that laser irradiation amplified the FK@IQ-4T1-induced antitumor immune response, the proliferation of CD8$^+$ CTLs was evaluated by Ki-67 immunofluorescence staining. Figure 5e showed that FK@IQ-4T1+L significantly induced the proliferation of tumor infiltrating CTLs (CD8$^+$/Ki-67$^+$), which was 2-fold

more efficient than FK@IQ-4T1. FK@IQ-4T1+L significantly increased the frequency of TNF-α and IFN-γ dual-positive effector CD8$^+$ T cells (Fig. 5f). The increased proliferation and activation of effector CTLs provides further evidence that FK@IQ-4T1+L treatment combining the presentation of tumor antigens and JQ1-mediated inhibition of the PD-L1/PD-1 blockade effectively activates TNF-α$^+$/IFN-γ$^+$ CTLs capable of killing tumor cells by secreting various cytokines including perforin, granzymes, and granulysin[43]. Compared to FK@IQ+L and FK@IQ-4T1, FK@IQ-4T1+L significantly increased the CD8$^+$ CTL to Treg ratio by 2.7-fold and 2.9-fold, respectively, indicating that FK@IQ-4T1+L significantly induced the anti-tumor immunity (Fig. 5g).

**PVAX blocks the recurrence and metastasis of autologous tumors.** It has been recently reported that neoantigen-loaded tumor vaccines are able to generate antigen-specific immunological responses for personalized immunotherapy[18]. To demonstrate the necessity of employing autologous tumor vaccines for cancer immunotherapy, tumor cells from 4T1 and EMT6 breast tumor xenografts were used as the autologous and allogeneic cell sources, respectively. The respective FK@IQ-4T1 and FK@IQ-EMT6 vaccines were then employed for cross-vaccination of EMT6 or 4T1 tumor-bearing BALB/c mice at the surgical bed of the primary tumors (Fig. 6a). We observed that FK@IQ-4T1 vaccination and laser irradiation completely eradicated the recurrent 4T1 tumors and suppressed the growth of the distant tumors. However, this combination failed to inhibit the growth of the EMT6 recurrent tumors (Fig. 6b, c and Supplementary Fig. 19). In parallel, vaccination with allogeneic FK@IQ-EMT6 vaccine showed inefficacy to inhibit the recurrence and establishment of distant 4T1 tumors, although it efficiently suppressed the establishment of distant EMT6 tumors (Fig. 6d, e). The efficacy of autologous tumor vaccines to target recurrent and distant tumors suggests the potential of PVAX for personalized immunotherapy.

To elucidate the mechanisms responsible for inducing the patient-specific antitumor immune response, the frequency of CD8$^+$ CTLs was measured in the distant tumors. Figure 6f, g displayed that FK@IQ-EMT6+L and FK@IQ-4T1+L both significantly promoted the intratumoral infiltration of CD8$^+$ CTLs in the homogenous tumors. In contrast, FK@IQ-4T1 vaccination and laser irradiation on the surgical bed of EMT6 tumors induced moderate recruitment of CD8$^+$ CTLs in the distant tumors, suggesting that the PVAX induced tumor-specific antitumor immunity. These observations confirm that the autologous tumor cell source is crucial for personalized cancer immunotherapy.

To further verify the tumor-specific T cell response induced by the hydrogel vaccine, the peripheral blood mononuclear lymphocytes (PMBCs) and spleen lymphocytes were collected from

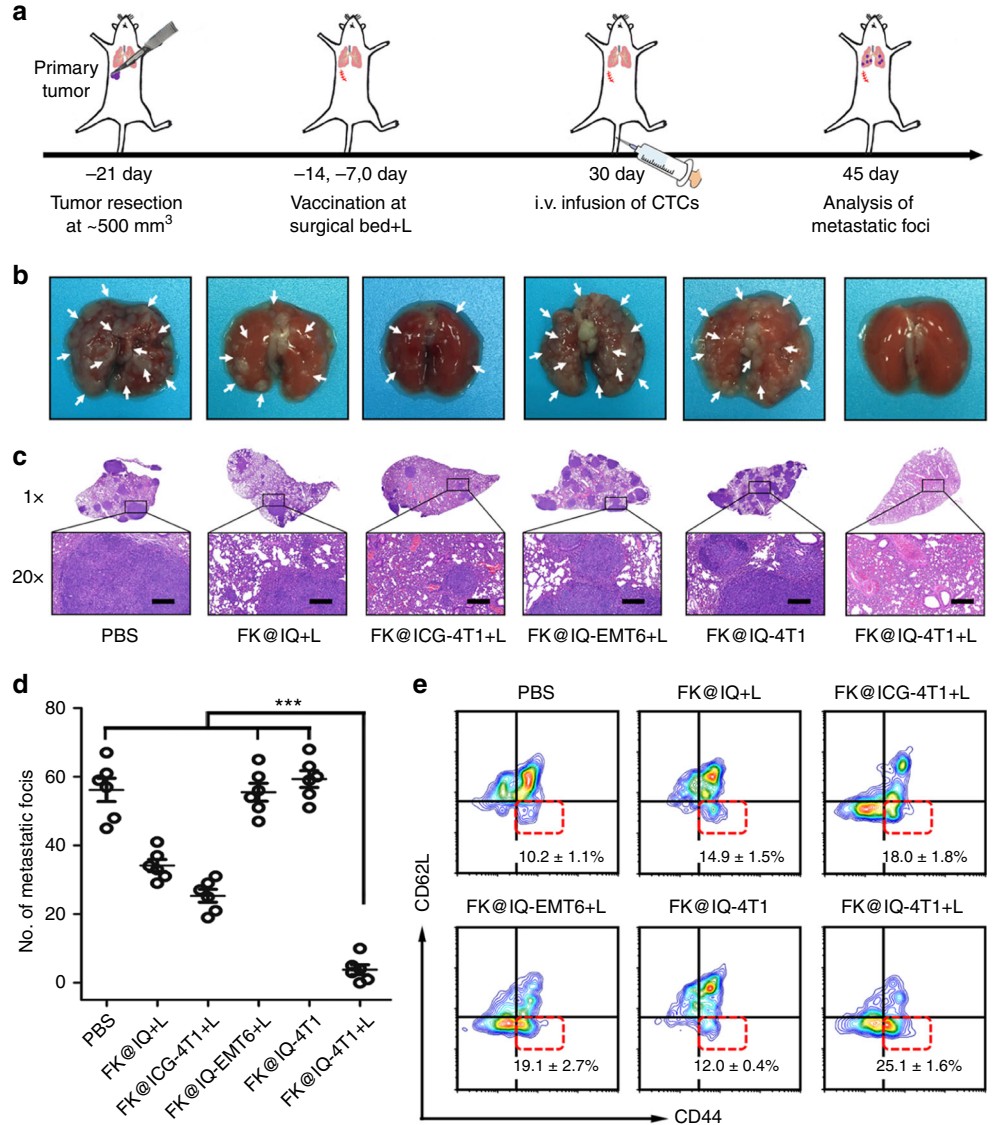

**Fig. 7** Metastasis prevention via PVAX-induced long term immune memory effects. **a** Therapeutic schedule for PVAX-mediated inhibition of tumor metastasis. **b** Representative photographs of lung tissue. **c** H&E staining of the lung tissue collected at day 45 (Scale bars = 200 μm). **d** Quantification of pulmonary metastasis nodes on mice pre-vaccinated with different vaccines. The mean metastasis nodes were analysed using one-way ANOVA. ***$p <$ 0.01. Data represent mean ± s.d. ($n = 6$). **e** Flow cytometry plots and proportions of effector memory T cells in the spleen (gated on CD8$^+$ CD11b$^+$) examined at the same day for i.v. infusion of the 4T1 cells

the BALB/c mice 3 days post vaccination. The PMBCs and spleen lymphocytes were co-cultured with 4T1, EMT6, or NIH 3T3 cells, respectively. The secretion of IFN-γ by the PMBCs and spleen lymphocytes was then examined by ELISA kits 48 h postincubation. Figure 6h, i demonstrated that the FK@IQ-4T1-vaccinated PMBCs and spleen lymphocytes secreted IFN-γ much more efficiently than FK@IQ-EMT6+L when co-cultured with the 4T1 cells. In contrast, the combination of FK@IQ-EMT6+L highly efficiently induced IFN-γ secretion by the PMBCs and spleen lymphocytes when incubated with the EMT6 cells. These results consistently verified that PVAX induced tumor-specific immune responses, which is essential for personalized immunotherapy.

**Long term immune-memory effects.** Encouraged by the high efficacy of the PVAX to induce a systemic immune response for inhibiting tumor recurrence and distant tumor metastasis, we

evaluated the potential of PVAX for preventing lung metastasis by inducing the immune memory effect. The immunological memory of the adaptive immune response is responsible for the priming and recognition of "old antigens", which is essential for preventing tumor relapse or metastasis post-surgery[20]. To mimic post-surgical tumor metastasis derived from CTCs, the surgical bed of the 4T1 tumors was vaccinated with FK@IQ-4T1 and treated with 808 nm laser triplicate at a time interval of 6 days. A lung metastasis tumor was then established by intravenously infusing 4T1 tumor cells into the BALB/c mice 30 days after the third vaccination (Fig. 7a). FK@IQ-EMT6+L and FK@IQ-4T1 all showed inefficacy to inhibit the establishment of metastatic foci in the lung. In contrast, FK@IQ-4T1+L dramatically suppressed the lung metastasis of 4T1 CTCs. The metastatic foci in the FK@IQ-4T1+L group were 14.4-fold less than those of the FK@IQ-4T1 groups (Fig. 7b–d).

To understand the mechanism underlying the tumor-specific anti-metastasis property of the PVAX, the effector memory

(CD11b$^+$CD8$^+$CD44$^+$CD62L$^-$) T cells in the spleen, LNs and bone marrow were examined by using flow cytometric measurement on the same day as the i.v. infusion of 4T1 cells. FK@IQ-4T1 vaccination and laser irradiation significantly increased the frequency of effector memory T cells in the spleen, bone marrows and LNs, whereas FK@IQ-4T1, FK@IQ+L and FK@ICG-4T1+L all showed moderate efficacy to generate effector memory T cells (Fig. 7e and Supplementary Fig. 20). In parallel, the mouse groups treated with the allogeneic tumor vaccine FK@IQ-EMT6 and laser irradiation displayed low efficacy to inhibit lung metastasis of the 4T1 tumor cells due to the lack of tumor-specific antigens. Hematoxylin-eosin (H&E) staining at the end of the antitumor study revealed no obvious histological damage in the major organs (e.g., heart, liver, spleen, lung, and kidney), supporting the biosafety of PVAX (Supplementary Fig. 21).

## Discussion

Tumor relapse and distant metastasis postoperation are lethal reasons for cancer-caused modality. In past years, immunotherapy has showed promise for clinical treatment of the recurrent or metastatic tumors. In comparison with the currently existing immunotherapeutic regimens, PVAX we demonstrated in this study is of several distinct advantages. First, PVAX was rationally designed for postoperative cancer immunotherapy by NIR light-triggered tumor antigen release and initiating tumor-specific immune response. PVAX was loaded with JQ1 for PD-L1 blockade. This is cost-efficient, convenient for preparation and easy for storage by avoiding the use of checkpoint antibody. Second, the autologous cancer vaccines contain unique tumor antigens encoded by gene mutations specific to that individual tumor. These antigens might be more immunogenic than commonly shared tumor antigens and result in stimulating effective and long-lasting anticancer responses in the patient. Moreover, although whole tumor cell vaccines had been extensively exploited in past years, their therapeutic efficacy was severely suppressed in the immunosuppressive tumoral microenvironment. In contrast, PVAX loaded with JQ1 can vigorously overcome immune evasion by blocking the PD-L1 checkpoint. Compared to neoantigens for cancer immunotherapy, PVAX can be readily prepared in a cost-efficient manner, which is highly desired by the late stage or metastatic cancer patients. Furthermore, the tissue penetration of light could be limited even with NIR lasers, the endoscope-based devices with laser optical fibers could be used for laser illumination of the tumors located deeply inside the body (e.g., colorectal tumors). The clinical translation of PVAX could thus be realistic.

In summary, we have demonstrated the efficacy of a tumor cell-derived cancer vaccine for personalized immunotherapy in this study. The vaccine was prepared by encapsulating JQ-1 and ICG co-loaded tumor cells within a hydrogel matrix for local injection and overcoming tumor burden. The vaccine can spatiotemporally release tumor-specific antigens and the BRD4 inhibitor JQ1 at the surgical bed in a NIR light-controlled manner, which can simultaneously elicit the antitumor immunity and block the PD-L1/PD-1 checkpoint to prevent tumor recurrence and metastasis. Moreover, NIR laser-triggered activation of PVAX elicited a sustained immune-memory effect, which effectively suppressed tumor metastasis for a period of 30 days after vaccination. The personalized tumor vaccine we proposed herein might offer a new strategy for post-surgical immunotherapy of the recurrent and metastatic tumors.

## Methods

**Materials**. 3-(4,5-dimethylthiazol-2-yl)-2,5-diphenyltetrazolium bromide (MTT), Hoechst 33342 and DAPI were obtained from Life Technologies (Shanghai, China). Collagenase IV, DNase, hyaluronidase (HAase) were purchased from Sigma-Aldrich (Shanghai, China). Tumor penetrable Fmoc-KRGDK (FK) peptide and a tumor non-penetrable analog Fmoc-KRGDC (FC) peptide were synthesized by solid phase

chemistry. Indocyanine Green (ICG) was purchased from Frontier Scientific Inc. (Logan, USA). JQ1 was ordered from Selleck Chemicals (USA). Coumarin 6 (C6) was obtained from J&K Chemicals (Shanghai, China). Anti-PD-L1 antibody was ordered from Abcam (Shanghai, China). ELISA kits for TNF-α, IFN-γ and IL-6 assay were all purchased from Neobioscience Co., Ltd. (Shenzhen, China). Phorbol-12-myristate-13-acetate (PMA) and Ionomycin were purchased from Dakewe Biotech Co., Ltd. (Shenzhen, China). All other regents were obtained from Sinopharm Chemical Reagent Co., Ltd. (Shanghai, China) and used without further purification.

**Cell lines**. NIH 3T3 mouse embryonic fibroblast cells, 4T1 murine breast tumor cells and RAW 264.7 macrophage cells were purchased from the Cell Bank of Chinese Academy of Science (Shanghai, China) and maintained in 10% v/v FBS-supplied RPMI 1640 medium. EMT6 murine breast tumor cells were obtained from ATCC and cultured in Waymouth's MB 752/1 Medium added with 10% v/v of FBS, 100 U/mL of penicillin and 100 μg/mL of streptomycin. The cells were incubated at 37 ℃ humidified environment with 5% CO$_2$ supply.

**Animals**. Four-week old BALB/c mice (18–20 g, female) were obtained from Shanghai Experimental Animal Center (Shanghai, China). All animal procedures were carried out under the guidelines approved by the Institutional Animal Care and Use Committee (IACUC) of the Shanghai Institute of Material Medica, Chinese Academy of Sciences.

**Photothermal effect and laser-triggered drug release in vitro**. The photothermal effects of the PVAX vaccines was firstly examined. Briefly, 30 μL of FK@IQ-4T1 suspension was diluted with PBS to the desired ICG concentrations. The solutions were then irradiated with an 808 nm laser at predetermined photo density for 2 min (Changchun New Industries Optoelectronics Tech. Co., Ltd, Changchun, China). Laser irradiation-induced temperature elevation was recorded using an IR camera (IRTech Co. Ltd., Shanghai, China).

To determine laser-triggered JQ1 release, 250 μL of FK@IQ-4T1 was added into a 1.5 mL EP tube and 250 μL of PBS buffer solution (pH 7.4) containing 0.5 wt% Tween-20 was added on the top of the hydrogel. The hydrogel suspension was then irradiated with an 808 nm laser at a selected photo density for 2 min with replacement of the supernatant solution (200 μL) at predetermined time intervals. All experiments were performed at 37 ℃. The JQ1 content in the collected solutions were then analyzed using HPLC. All experiments were performed in triplicate.

To evaluate laser-induced release of the tumor antigens, the collected solutions were centrifuged at 10,000 rpm for 10 min to remove the supernatant. After adding gel loading buffer, the precipitate was boiled at 100 ℃ for 5 min and then electrophoresed on 8% SDS-PAGE. The full-size image was shown in Supplementary Figure 22.

**NIR light-promoted intratumoral penetration of C6 in vitro**. NIR light-triggered intratumoral JQ1 release from PVAX vaccine was first evaluated using a multicellular spheroid (MCSs) tumor model in vitro. Briefly, a 48-well plate was pre-coated with 2.0 % (w/v) agarose before seeding with 4T1 cells ($2 \times 10^3$ cells/well). The cell culture medium was refreshed after 7 days incubation. Each well was added with a cell culture insert with 8 μm pore size containing 150 μL of IC-4T1, FC@IC-4T1 or FK@IC-4T1 suspension. The insert was then illuminated with 808 nm laser for 2 min at photo density of 2.0 W/cm$^2$. The MCSs were continuously cultured for 12 h before subjected to flow cytometry using a FACS Calibur system (BD Biosciences, Oxford, UK) and confocal laser scanning microscopy (CLSM) (Carl Zeiss LSM710, Germany).

**Biocompatibility of the peptide hydrogels**. RAW 264.7 and BMDCs cells were seeded into 96-well culture plates at a density of $3 \times 10^3$ cells per well for 24 h. The cells were then incubated with the FK or FC hydrogels for 48 h at the desired concentrations. The cell viability was then examined using the sulforhodamine B (SRB) assay.

**PD-L1 inhibition with JQ1**. To investigate JQ1-induced downregulation of PD-L1 on the membrane of 4T1 cells, the cells were seeded in 6-well culture plates at a density of $3 \times 10^4$ cells/well and allowed to attach overnight. The cells were then treated with 20 nM of IFN-γ for 24 h to simulate the expression of PD-L1. Afterwards, the cells were incubated with free or released JQ1 for 72 h at a concentration of 200 nM. PD-L1 cell surface expression on 4T1 cells was then examined using flow cytometry.

**Photothermal effect in vivo**. To examine the photothermal effects of the vaccines in vivo, FK@IQ-4T1 was subcutaneously injected into the right flank of BALB/c mice at a ICG concentration of 2.5 mg/kg. The injection sites were then irradiated with 808 nm laser for 2 min at predetermined photo densities. Laser irradiation induced photothermal effect was recorded with an IR camera.

**PVAX-induced immune response in vitro and in vivo**. To investigate the immunological effects of PVAX, BMDCs were separated from BALB/c mice. BMDCs

(1 × 10⁵ cells) were seeded into 24-well culture plates and allowed to attach overnight. Vaccines were added into a cell culture insert with 8 μm pore size and immersed in the 24-well culture plates. The cells were harvested 24 h post vaccination, co-stained with anti-CD11c-FITC (eBioscience, Clone: N418, Catalog: 11-0114-85), anti-CD80-PE (eBioscience, Clone: 16-10A1, Catalog: 12-0801-85) and anti-CD86-APC (eBioscience, Clone: GL1, Catalog: 17-0862) antibodies according to the procedure of the manufacturer, and then examined by flow cytometry.

For DC maturation examination in vivo, BALB/c mice were treated with varying vaccine formulations. The inguinal lymph nodes were collected 3 days post vaccination to collect lymphocytes. The frequency of DC maturation in the lymph nodes was then examined by flow cytometry after immunofluorescence staining with anti-CD11c-FITC (eBioscience, Clone: N418, Catalog: 11-0114-85), anti-CD80-PE (eBioscience, Clone: 16-10A1, Catalog: 12-0801-85) and anti-CD86-APC (eBioscience, Clone: GL1, Catalog: 17-0862) antibodies according to the procedure of the manufacturer.

To verify vaccination-induced immune response, serum samples were collected at predetermined time points from treated mice. The serum concentrations of TNF-α, IFN-γ, and IL-6 were analyzed with ELISA kits according to the manufacturer's protocols using samples collected at 1, 3, or 7 days post vaccination and laser irradiation.

To determine the serum proportion of CTLs post vaccination, mouse blood was collected 3 days post vaccination. Blood was then centrifuged for 10 min at 1000 rpm to remove serum. Blood cells were incubated with red blood cell lysis buffer for 3 min at 4 °C and then centrifuged for 10 min at 1000 rpm to collect lymphocytes. The proportions of CD4⁺ or CD8⁺ T cells were examined using flow cytometry after staining with anti-CD3-PerCP-Cy5.5 (eBioscience, Clone: 145-2C11, Catalog: 45-0031), anti-CD4-FITC (eBioscience, Clone: GK1.5, Catalog: 11-0041) and anti-CD8-PE (eBioscience, Clone: 53–6.7, Catalog: 12-0081) antibodies according to the procedure of the manufacturer.

**Tumor penetration of hydrogel vaccines in vivo**. The tumor penetration profile of FK hydrogel in vivo was examined in 4T1 tumor bearing BALB/c mice. Briefly, 4T1 tumor-bearing mice were intratumorally injected with 100 μL of ICG, FC@IC-4T1 or FK@IC-4T1 at an identical ICG concentration of 2.5 mg/kg. The intratumoral accumulation and retention of the hydrogel vaccine was then examined by fluorescence imaging using an IVIS animal imaging system at predetermined time points post-injection (Ex = 710 nm, Em = 780 nm for ICG).

To examine the tumor penetration ability of the hydrogel vaccines in vivo, 100 μL of IC-4T1, FC@IC-4T1 or FK@IC-4T1 suspension was intratumorally injected into the 4T1 tumor-bearing mouse at an identical ICG or C6 dose of 2.5 mg/kg. Eight hours post-injection, the tumors were irradiated with 808 nm laser for 2 min at a photo density of 2.0 W/cm². The tumors were harvested, frozen sectioned and examined using CLSM measurement (C6 channel, Ex = 488 nm; ICG channel, Ex = 633 nm).

**Anti-tumor study using autologous tumor model**. To evaluate the antitumor effect of PVAX in vivo, 5 × 10⁵ 4T1 cells were transplanted into the right mammary fat pad of BALB/c mice. The tumors were surgically resected when the tumor volume reached 500 mm³. The tumor excisions were digested and used for vaccine preparation. The mice were then randomly divided into six groups (n = 6). The mouse groups were intratumorally injected with the desired formulations and selectively irradiated with 808 nm laser for 2 min at a photo density of 2.0 W/cm² (i.e., 1: PBS; 2: FK@IQ +L; 3: FK@ICG-4T1+L; 4: FK@IQ-4T1 and 5: FK@IQ-4T1+L). The treatment was repeated three times at a time interval of 6 days. After the first treatment, 1 × 10⁵ 4T1 tumor cells were transplanted into the left flank of each mouse to establish the distant metastasis tumor model. Body weight and tumor growth were monitored every 2 days., all the major organs (i.e., heart, liver, spleen, lung, and kidney) were collected at the end of the anti-tumor study and examined by H&E staining. The tumors were harvested, fixed and sliced for H&E, Ki-67, and TUNEL staining.

To examine the antitumor immune response, BALB/c mice were inoculated with 5 × 10⁵ 4T1 cells and divided into six groups by receiving relevant treatment abovementioned. The treatments were repeated a total of three times at an interval of 6 days. The tumors were harvested 3 days after each treatment, digested with collagenase IV, hyaluronidase and DNase for 30 min at 37 °C. After passage through 75 μm filters, the monodispersed tumor infiltrating lymphocytes (TILs) were enriched using lymphocyte separation medium. The collected TILs were incubated with anti-CD3-PerCP-Cy5.5 (eBioscience, Clone: 145-2C11, Catalog: 45-0031), anti-CD4-FITC (eBioscience, Clone: GK1.5, Catalog: 11-0041) and anti-CD8-PE (eBioscience, Clone: 53-6.7, Catalog: 12-0081) antibodies according to the standard protocols to determine the intratumoral infiltration of CD4⁺ or CD8⁺ T cells using a flow cytometry. In parallel, the frequency of effector CD8⁺ T cells and their proliferation were further investigated by flow cytometry examination after co-staining with anti-Ki-67- Alexa 488 (Cell Signaling Technology, Clone: D3B5, Catalog: 118825), anti-IFN-γ-FITC (BD Pharmingen, Clone: XMG1.2, Catalog: 554411) and anti-TNF-α-Alexa 647 (BD Pharmingen, Clone: MP6-XT22, Catalog: 557730) antibodies antibodies according to the standard protocols. The frequency of regulatory T cells was examined by staining the TILs with anti-CD3-PerCP-Cy5.5 (eBioscience, Clone: 145-2C11, Catalog: 45-0031), anti-CD4-FITC (eBioscience, Clone: GK1.5, Catalog: 11-0041), anti-CD25-APC (eBioscience, Clone: PC61.5, Catalog: 17-0251), and anti-Foxp3-PE (eBioscience, Clone: NRRF-30, Catalog: 12-4771) antibodies according to the standard protocols. To

investigate the immune memory effects, lymphocytes from bone marrow were separated 30 days following the final treatment and stained with anti-CD8-FITC (eBioscience, Clone: 53-6.7, Catalog: 11-0081-82), anti-CD11b-APC (eBioscience, Clone: M1/70, Catalog: 17-0112), anti-CD44-PerCP-Cy5.5 (eBioscience, Clone: IM7, Catalog: 45-0441), and anti-CD62L-PE (eBioscience, Clone: MEL-14, Catalog: 12-0621) antibodies according to the standard protocols before analyzed using flow cytometry. All experiments were performed in triplicate.

**Anti-tumor study using an allogeneic tumor model**. To demonstrate the potential of PVAX for personalized immunotherapy, the antitumor study was performed using an allogeneic tumor model. Briefly, 4T1 and EMT6 tumor xenografts were established by subcutaneously injecting 4T1 or EMT6 cells into the right mammary fat pad of BALB/c mice. The mice were then randomly divided into six groups (n = 6). The tumors were surgically resected when the tumor volume reached 500 mm³. The tumor excisions were digested and used for preparation of the tumor cell vaccines. The hydrogel vaccines were intratumorally injected into the surgery bed and selectively irradiated with 808 nm laser for 2 min at a photo density of 2.0 W/cm² when the residual tumors grew to 100 mm³ (i.e., 1: PBS; 2: FK@IQ4T1+L; 3: FK@IQ-EMT6+L). The treatment was repeated for three times at a time interval of 6 days. The intratumoral infiltration of CD4⁺ and CD8⁺ T cells in the secondary tumors was examined by flow cytometry.

PVAX-induced tumor-specific immune response was examined ex-vivo according to a literature reports with slight adaption[44, 45]. Briefly, peripheral blood mononuclear cells (PBMCs) and spleen lymphocytes were separated from BALB/c mice 3 days post vaccinations. PBMCs or spleen lymphocytes (1 × 10⁵ cells) were then co-cultured with 4T1, EMT6, NIH 3T3 cells (2 × 10⁵ cells) or Phorbol-12-myristate-13-acetate (PMA) + Ionomycin (PMA 0.5 μg/mL, Ionomycin 10 μg/mL) for 48 h, respectively. The culture medium were then collected to determine the secretion of IFN-γ by using ELISA kits according to the manufacturer's protocols.

**Anti-metastasis effect of PVAX**. To investigate the long term immune memory effects of PVAX, 4T1 cells (5 × 10⁵) were first transplanted into the mammary fat pad of mice. When the tumor volume reached 500 mm³, the tumors were surgically resected to collect the autologous tumor excisions for cancer vaccine fabrication. Secondly, BALB/c mouse were randomly divided into six groups (n = 6 in each group). The mice in each group were subcutaneously injected with the desired formulations and selectively irradiated with 808 nm laser (i.e., 1: PBS; 2: FK@IQ+L; 3: FK@ICG-4T1+L; 4: FK@IQ-EMT6+L; 5: FK@IQ-4T1; 6: FK@IQ-4T1+L). The treatment was repeated for three times at a time interval of 6 days. To establish the lung metastasis tumor model, 30 days post the final vaccination, each mouse was intravenously infused with 1 × 10⁵ 4T1 tumor cells. The mice were sacrificed 15 days post tumor cell injection. The establishment of 4T1 tumors in the lung was examined by H&E staining.

**Statistical analysis**. All the data were presented as mean ± s.d.. Unpaired student's t-test (two-tailed) was used for comparison between two groups. One-way analysis of variance (ANOVA) was used for multiple-group analysis. Statistical significance was set at **p < 0.05, ***p < 0.01.

**Data availability**. All relevant data are available from the authors.

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

## Acknowledgements
The authors thank Prof. Ling Yu from Fudan University for assistance with rheology measurements. Financial supports from the National Natural Science Foundation of China (31671024, 31622025, 81521005 and 81690265) are gratefully acknowledged. All animal procedures were carried out under the guidelines approved by the Institutional Animal Care and Use Committee (IACUC) of the Shanghai Institute of Materia Medica, Chinese Academy of Sciences.

## Author contributions
T.W., D.W. and H.Y. conceived and designed the project. T.W., D.W., B.F., F.Z., H.Z. L. Z. and S.J. performed the experiments. T.W., D.W., H.Y. and Y.L. analyzed the data and wrote the manuscript. All the authors revised the manuscript and approved the submission.

## Additional information

**Competing interests:** The authors declare no competing interests.

