## [Peer Review File · Nature Communications]

Reviewers' comments:

Reviewer #1 (Remarks to the Author):

This manuscript by Wang et al. reported on a new strategy for cancer vaccine-mediated postoperative immunotherapy, based on the coencapsulation of the JQ1 and a photosensitizer ICG together with inactivated tumor cells into a hydrogel matrix for cancer immunotherapy. They then used intratumoral injection of these hydrogel followed by NIR laser irradiation to trigger tumor antigen and JQ1 release for inhibiting primary tumors and induce an immune response that attacked distal metastases. Some issues should be addressed before publication.

1. The authors removed majority of the primary tumor but the hydrogel was then implanted into surgical bed 7 days later when the recurrent tumors reached 100 mm^3 . It would be interesting to test if the vaccine hydrogels inhibit the local recurrent tumor when they are administered in other place of the body, as the authors also demonstrated that a systemic immune response was induced to inhibit distant tumor metastasis and the therapy showed impressive results.
2. The authors could also look at the status of DCs in treated tumors following the treatments, rather than only in draining lymph nodes in Figure 3d. It is unclear if DCs in the tumors can be activated by the heat and irradiation, as the local heat can cause the release of inflammatory cytokines and HSPs that activate the immune system.
3. The T cell infiltrate analyses should be reported as raw cell counts per mass of tumor, as percentages of cells can be misleading since the frequencies of many cell types in the tumors may be changing simultaneously following treatment.
4. The authors should clarify the gating strategy used in fig 4f, they stated cells were gated on CD3, but many cells appear CD4CD8 in each flow plot.
5. Cancer cells used for the development of vaccines contain a very high proportion of targets which are not cancer cell-specific. No evaluation of tumor-specific T cell responses was presented. The authors need to determine whether the tumor cells in the hydrogel was promoting 4T1-specific immunity. The study provided very limited mechanistic insights into the immunity elicited by hydrogel vaccination.
6. Tregs should be stated FACS data which were not shown. Tregs may be important as the inactivated whole-cell vaccines contain a lot of normal antigens, which may induce the immune tolerance.

Reviewer #2 (Remarks to the Author):

In the manuscript titled "Personalized Cancer Vaccine Immune Checkpoint Blockade for Effective Immunotherapy of Recurrent and Metastatic Tumors," Wang and associates present their extensive work on a whole tumor cell based therapy co-encompassed in FK thermosensitive hydrogels with a previously reported inhibitor of PD-L1 expression. The authors' extensive background and expertise on the subject matter is evident from their previous work in the field

and by the vast degree of highly relevant included supporting information with the submitted manuscript. Some relevant questions remain:

1. The FK and FC hydrogels are assembled from Fmoc-KCRGDK and Fmoc-FCRGDC as described in the text. However, the peptides are not only N-terminal Fmoc protected but have also coupled Fmoc to the side chain of the N-terminal lysine side chain. Please comment on the design of peptide and the rationale for the protecting groups. (i.e. changes in the physical properties of the FK peptide assemblies with and without Fmoc).
2. Previous work using the RGD tumor penetrating peptide motif utilized it to deliver a covalently bound cargo to the tumor. The 4T1 cells, or in this case the lysate created after NIR irradiation, is not bound to the Nrp-1 binding peptide. What do the authors propose would be the driving mechanism to increase antigen uptake and presentation of the tumor lysate in vivo?
3. With respect to clinical translation, the authors utilized intratumoral injections in the paper and then describe the potential for clinical application using far reaching fiber optics for invasive tumors, assuming that the vaccine can be delivered to the tumor. A critical experiment to highlight clinical application in these cases would involve administration of the vaccine to an accessible site, including distant to the tumor. This is easily testable by local (peritumoral) & distant injection of the vaccine after tumor resection using the same timeline in Fig4 or Fig5. Even if the data is negative it is of interest to show the importance of the route of administration.
4. Supplemental Figure 9. The text (line 174) describes enhanced retention of ICG with FK hydrogels compared to FC. Without quantification (photon/s) this is not justified, in fact the retention of ICG from IC-4T1 alone is quite surprising. Please revise your conclusions or the figure to support it.
5. The authors show the ability of the vaccine to prevent tumor reoccurrence and metastatic disease after resection of the primary tumor. Please comment or provide data on efficacy against the primary tumor with PVAX made from biopsied cells or the implanted tumor cell line prior to resection.
6. Figure 4 and 5, the authors show tumor size and survival out to ~day 25. Do the tumors grow out eventually? If so, does additional administration of booster FK@IQ-4T1 or even FK@IQ doses help prevent tumor growth? We see many vaccine candidates that only delay tumor reoccurrence for a period of time. Insight into the long term efficacy is highly relevant.
7. Figure 6e. The authors compare effector memory cells described as (CD11b+CD8+CD44+CD62Llow), from isolated bone marrow. Did the authors not look in the draining lymph nodes or spleen for recently activated effector memory cells? Please provide rationale for only showing data from or investigating bone marrow.

Minor comments

1. Line 241: 2.5 is missing units
2. Methods are missing details on statistical analysis
3. Figure 5 and S18, the figures would be improved by keeping the colors for the groups consistent.
4. Line 369: 'narrow' should be 'marrow'
5. Line 372: change activate to generate.
6. Line 383: 'frist' spelling error.
7. Line 48: remove despite or reword sentence for grammar
8. Line 50: current standard of care does not increase morbidity, if it did it would not be the standard. Chemotherapy and radiation can certainly decrease quality of life however.

Response to the Reviewers

Reviewer #1:

This manuscript by Wang et al. reported on a new strategy for cancer vaccine-mediated postoperative immunotherapy, based on the co-encapsulation of the JQ1 and a photosensitizer ICG together with inactivated tumor cells into a hydrogel matrix for cancer immunotherapy. They then used intratumoral injection of these hydrogel followed by NIR laser irradiation to trigger tumor antigen and JQ1 release for inhibiting primary tumors and induce an immune response that attacked distal metastases. Some issues should be addressed before publication.

1. The authors removed majority of the primary tumor but the hydrogel was then implanted into surgical bed 7 days later when the recurrent tumors reached 100 mm³. It would be interesting to test if the vaccine hydrogels inhibit the local recurrent tumor when they are administered in other place of the body, as the authors also demonstrated that a systemic immune response was induced to inhibit distant tumor metastasis and the therapy showed impressive results.

Response: We appreciate the critical comment of the reviewer.

According to the comment of the reviewer, we investigated the therapeutic efficacy of the vaccine hydrogel by subcutaneous (s.c) injection at the distant site. The results showed that FK@IQ-4T1 + L treatment at the distant site moderately suppressed the recurrence of 4T1 primary tumor (Supplementary Fig. 17a-c), suggesting tumor-specific delivery of JQ-1 is critical to suppress PD-L1-mediated immune evasion and inhibit tumor recurrence.

Interestingly, FK@IQ-4T1 + L at the distant site highly efficiently inhibited the growth of the distant tumor, which could be attributed to vaccine-induced systemic immune response since the systemic antitumor immunity is independent on the injection site (Supplementary Fig. 17d,e).

We added above discussion in Page 15 Line 286-294 of the revised manuscript.

Supplementary Fig. 17. PVAX-performed immunotherapy in the recurrent and metastatic 4T1 tumor xenograft model treated at the distant sites. (a) Treatment schedule for the immunotherapy; (b) Average tumor growth curves of the recurrent tumors of 4T1 tumor-bearing mice receiving different treatments at the distant sites, laser irradiation for 2 min at photodensity of 2.0 W/cm²; (c) Survival percentage of the 4T1 tumor-bearing mice treated by surgical resection and PVAX-performed immunotherapy at the distant sites; (d) Average tumor growth curves of the rechallenged tumors receiving different treatments at the distant sites, laser irradiation for 2 min at photodensity of 2.0 W/cm²; (e) Tumor-free percentages of the distant tumor rechallenged mice (*** p < 0.01).

2. The authors could also look at the status of DCs in treated tumors following the treatments, rather than only in draining lymph nodes in Figure 3d. It is unclear if DCs in the tumors can be activated by the heat and irradiation, as the local heat can cause the release of inflammatory cytokines and HSPs that activate the immune system.

Response: Figure 3d in the submitted manuscript displayed that vaccination significantly accelerated DC maturation in the LNs of the tumor-free BALB/c mice.

According to the suggestion of the reviewer, we examined the laser-promoted intratumoral DC maturation of the tumor xenograft-bearing BALB/c mice. Fig. 5a in the revised

manuscript showed that FK@IQ-4T1+L significantly accelerated DC maturation in the tumor.

Fig. 5. Vaccine-induced antitumor immunity *in vivo*. (a) The hydrogel vaccines accelerated DC maturation (CD11c⁺CD80⁺CD86⁺) in the tumor examined 3 days post treatment.

3. The T cell infiltrate analyses should be reported as raw cell counts per mass of tumor, as percentages of cells can be misleading since the frequencies of many cell types in the tumors may be changing simultaneously following treatment.

Response: The T cell infiltration in the tumor was normalized with the tumor mass as showed in Fig. 5c in the revised manuscript.

Fig. 5. (c) Tumor mass-normalized intratumoral infiltration of CD8⁺ T cells in the recurrent tumors examined 10 days post treatment (***) p < 0.01).

4. The authors should clarify the gating strategy used in fig 4f, they stated cells were gated on CD3, but many cells appear CD4⁻CD8⁻ in each flow plot.

Response: We appreciated the insightful comments of the reviewer.

The enriched T lymphocytes in the tumor were not gated on CD3⁺ T cells as showed in Fig. 4f of the submitted manuscript. Therefore, the majority of the cell population was CD4⁻CD8⁻ in the flow plot.

The T lymphocyte cells in the tumor were gated on CD3⁺ T cells as showed in Fig. 5b of the revised manuscript.

Fig. 5. (b) Flow cytometric examination of the intratumoral infiltration of CD4⁺ and CD8⁺ T cells (gated on CD3⁺ T cells).

5. Cancer cells used for the development of vaccines contain a very high proportion of targets which are not cancer cell-specific. No evaluation of tumor-specific T cell responses was presented. The authors need to determine whether the tumor cells in the hydrogel was promoting 4T1-specific immunity. The study provided very limited mechanistic insights into the immunity elicited by hydrogel vaccination.

Response: We appreciated the critical comments of the reviewer.

To investigate whether the hydrogel vaccine induced tumor-specific T cell response, the peripheral blood mononuclear lymphocytes (PMBCs) and spleen lymphocytes were collected from the BALB/c mice 3 days post vaccination. The PMBCs and spleen lymphocytes were co-cultured with 4T1, EMT6 or NIH 3T3 cells, respectively. The secretion of IFN- γ by the PMBCs and spleen lymphocytes was then examined by ELISA kits 48 h post-incubation. Fig. 6h,i demonstrated that the FK@IQ-4T1-vaccinated PMBCs and spleen lymphocytes secreted IFN- γ much more efficiently than FK@IQ-EMT6+L when co-cultured with the 4T1 cells. In contrast, the combination of FK@IQ-EMT6+L highly efficiently induced IFN- γ secretion by the PMBCs and spleen lymphocytes when incubated with the EMT6 cells. These results consistently verified that PVAX induced tumor-specific immune responses, which is essential for personalized immunotherapy.

We added above discussion and the method in Page 20 Line 378-389 of the revised manuscript.

Fig. 6. (h, i) Vaccination-induced secretion of tumor specific IFN- γ by the (h) PMBCs, and (i) the spleen lymphocytes, respectively.

6. Tregs should be stated FACS data which were not shown. Tregs may be important as the inactivated whole-cell vaccines contain a lot of normal antigens, which may induce the immune tolerance.

Response: We appreciate the suggestion of the reviewer.

Intratumoral infiltration of Tregs was examined by using flow cytometric measurement. The FACS data of Tregs and the CD8⁺/Treg ratios were showed in Fig. 5c and Fig. 5g, respectively in the revised manuscript. Fig. 5c demonstrated that the Treg frequency in the vaccinated groups (*e.g.*, FK@IQ-4T1 or FK@IQ-4T1+L) was much lower than that of the PBS group, verifying vaccination by FK@IQ-4T1+L significantly suppressed the immune tolerance of the recurrent tumor. Moreover, in comparison with FK@IQ+L and FK@IQ-4T1, FK@IQ-4T1+L significantly increased the CD8⁺ CTL to Treg ratio by 2.7- and 2.9-fold, respectively, indicating the combination of FK@IQ-4T1 vaccination with laser irradiation was crucial to enhance the antitumor immune response (Fig. 5g).

Fig. 5. (c) The frequency of Tregs in the recurrent tumors after different treatments examined 10 days post treatment; (g) Ratios of tumor-infiltrating CD8⁺ T cells Vs. Tregs in the recurrent tumor examined 10 days post the first treatment (***) p < 0.01).

Reviewer #2

In the manuscript titled “Personalized Cancer Vaccine Immune Checkpoint Blockade for Effective Immunotherapy of Recurrent and Metastatic Tumors,” Wang and associates present their extensive work on a whole tumor cell based therapy co-encompassed in FK thermosensitive hydrogels with a previously reported inhibitor of PD-L1 expression. The authors’ extensive background and expertise on the subject matter is evident from their previous work in the field and by the vast degree of highly relevant included supporting information with the submitted manuscript.

Response: We appreciate the encouraging comments of the reviewer.

1. The FK and FC hydrogels are assembled from Fmoc-KCRGDK and Fmoc-FCRGDC as described in the text. However, the peptides are not only N-terminal Fmoc protected but have also coupled Fmoc to the side chain of the N-terminal lysine side chain. Please comment on the design of peptide and the rationale for the protecting groups. (i.e. changes in the physical properties of the FK peptide assemblies with and without Fmoc).

Response: Two Fmoc groups were grafted on the N-terminal of the peptide sequence and the N-terminal lysine side chain to stabilize the self-assembled peptide nanoparticles and nanofibers via π - π interaction. Both the Fmoc groups were modified on the N-terminal of the peptide sequences, which would not interrupt the tumor penetration activity of the CRGDK motif.

We added above discussion in Page 6 Line 108-115 of the revised manuscript.

2. Previous work using the RGD tumor penetrating peptide motif utilized it to deliver a covalently bound cargo to the tumor. The 4T1 cells, or in this case the lysate created after NIR irradiation, is not bound to the Nrp-1 binding peptide. What do the authors propose would be the driving mechanism to increase antigen uptake and presentation of the tumor lysate in vivo?

Response: Exactly, covalently-conjugated RGD peptide enhanced tumor penetration of the cargo as reported in the literature reports including ours (Refs 31,32 in the revised manuscript). Meanwhile, Erkki Ruoslahti *et al* reported that combined administration of iRGD peptide improved the tumor penetration of small molecular chemotherapeutics (*i.e.*,

Doxorubicin) or nanoparticles by binding with Nrp-1 on the surface of the tumor cells and triggering an active transport process (Refs 33,38 in the revised manuscript).

We added above explanation in Page 10 Line 185-188 and cited a Nat. Commun. paper as reference 38 in the revised manuscript.

Reference 38. Liu, X. et al. In vivo cation exchange in quantum dots for tumor-specific imaging. Nat Commun. 8, 343 (2017).

3. With respect to clinical translation, the authors utilized intratumoral injections in the paper and then describe the potential for clinical application using far reaching fiber optics for invasive tumors, assuming that the vaccine can be delivered to the tumor. A critical experiment to highlight clinical application in these cases would involve administration of the vaccine to an accessible site, including distant to the tumor. This is easily testable by local (peritumoral) & distant injection of the vaccine after tumor resection using the same timeline in Fig 4 or Fig 5. Even if the data is negative it is of interest to show the importance of the route of administration.

Response: We appreciated the insightful comments of the reviewer.

We checked the therapeutic efficacy of the vaccine hydrogel by injecting the hydrogel at the distant sites. The results showed that FK@IQ-4T1+L at the distant site moderately inhibited 4T1 tumor recurrence (Supplementary Fig. 17a-c), suggesting vaccination at the surgical bed is critical for tumor-specific JQ-1 delivery and overcoming PD-L1-mediated immune evasion. In contrast, FK@IQ-4T1+L at the distant site completely suppressed distant tumor growth by inducing systemic immune response (Supplementary Fig. 17d,e).

We added the new data and above discussion in Page 15 Line 286-294 of the revised manuscript.

Supplementary Fig. 17. PVAX-performed immunotherapy in the recurrent and metastatic 4T1 tumor xenograft model at the distant sites. (a) Treatment schedule for the immunotherapy; (b) Average tumor growth curves of the recurrent tumors of 4T1 tumor-bearing mice receiving different treatments at the distant sites, laser irradiation for 2 min at photodensity of 2.0 W/cm²; (c) Survival percentage of the 4T1 tumor-bearing mice treated by surgical resection and PVAX-performed immunotherapy at the distant sites; (d) Average tumor growth curves of the rechallenged tumors receiving different treatments at the distant sites, laser irradiation for 2 min at photodensity of 2.0 W/cm²; (e) Tumor-free percentages of the distant tumor rechallenged mice (***) p < 0.01).

4. Supplemental Figure 9. The text (line 174) describes enhanced retention of ICG with FK hydrogels compared to FC. Without quantification (photon/s) this is not justified, in fact the retention of ICG from IC-4T1 alone is quite surprising. Please revise your conclusions or the figure to support it.

Response: We appreciate the critical comments of the reviewer.

We double-checked the intratumoral fluorescence intensity of ICG by photon quantification. FC@IC-4T1 and FK@IC-4T1 showed comparable ICG fluorescence intensity throughout the experimental period (Supplementary Fig. 9). However, the fluorescence intensity of the IC-4T1 group was 2.8- and 2.4-fold lower than that of the FC@IC-4T1 and FK@IC-4T1

respectively, when examined at 120 h postinjection. This could be explained by quick blood clearance of ICG in the IC-4T1 group. Therefore, it seems that FC and FK hydrogels both elongated the tumor retention of ICG. However, FK hydrogel enhanced the tumor penetration of ICG as showed Supplementary Fig. 10.

We added above discussion in Page 10 Line 178-185 of the revised manuscript.

Supplementary Fig. 9. (a) Fluorescence examination of the intratumoral accumulation and retention of the intratumorally injected cancer vaccines *in vivo*; (b) Normalized ICG fluorescence intensity of intratumorally injected FK@IC-4T1 vaccine as a function of injection time. The fluorescence imaging *in vivo* was performed using ICG (Ex = 710 nm, Em = 780 nm).

5. The authors show the ability of the vaccine to prevent tumor recurrence and metastatic disease after resection of the primary tumor. Please comment or provide data on efficacy against the primary tumor with PVAX made from biopsied cells or the implanted tumor cell line prior to resection.

Response: As suggested by the reviewer, we prepared the PVAX vaccines by using the implanted tumor cell line prior to tumor resection and testified their ability to regress the primary tumors.

[Redacted]

[We found] that the vaccine prepared from 4T1 cell line highly efficiently inhibited 4T1 tumor recurrence since the tumor cell line and the tumor cells extracted from the tumor xenograft share the same mutation, which can induce tumor-specific immune response by releasing tumor-specific antigens.

6. Figure 4 and 5, the authors show tumor size and survival out to ~day 25. Do the tumors grow out eventually? If so, does additional administration of booster FK@IQ-4T1 or even FK@IQ doses help prevent tumor growth? We see many vaccine candidates that only delay tumor recurrence for a period of time. Insight into the long term efficacy is highly relevant.

Response: We examined the anti-tumor performance of the hydrogel vaccine for a largely extended time period of 59 days. **Fig. 6b,d** in the revised manuscript showed that FK@IQ-4T1+L and FK@IQ-EMT6+L completely regressed the 4T1 and EMT6 tumor respectively, in the whole experimental period. More importantly, FK@IQ-4T1+L completely eradicated the 4T1 distant tumors, and FK@IQ-EMT6+L inhibited 60% of EMT6 distant tumor growth, verifying the vaccine hydrogel induced long term immune memory effect (**Fig. 6c,e**).

Fig. 6b-e. (b) Tumor growth curve of the recurrent EMT6 tumors, and (c) Tumor-free percentages of the distant EMT6 tumor-bearing BALB/c mice; (d) Distant tumor growth curves, and (e) Tumor-free percentages of the distant 4T1 tumor-bearing BALB/c mice.

7. Figure 6e. The authors compare effector memory cells described as (CD11b+CD8+CD44+CD62Llow), from isolated bone marrow. Did the authors not look in the draining lymph nodes or spleen for recently activated effector memory cells? Please provide rationale for only showing data from or investigating bone marrow.

Response: The frequency of effector memory cells in the spleen, bone marrow and draining lymph nodes (LNs) were examined by flow cytometric examination. The flow cytometry plots and proportions of effector memory T cells in the spleen, bone marrows (BM) and LNs were included in Fig. 7e, the Supplementary Fig. S20a and Fig. 20b in the revised manuscript.

Fig. 7. (e) Flow cytometry plots and proportions of effector memory T cells in the spleen (gated on $CD8^+CD11b^+$) examined at the same day for i.v. infusion of the 4T1 cells.

Supplementary Fig. 20. (a, b) Flow cytometry plots and proportions of effector memory T cells in the (a) bone marrow, and (b) the LNs (gated on $CD8^+CD11b^+$) examined at the same day for i.v. infusion of the 4T1 cells.

Minor comments

- Line 241: 2.5 is missing units

Response: A dosage unit of “mg/kg” was added after 2.5.

2. Methods are missing details on statistical analysis

Response: The statistical methods were added in the revised manuscript.

3. Figure 5 and S18, the figures would be improved by keeping the colors for the groups consistent.

Response: The color for all the curves was kept consistent in the revised manuscript.

4. Line 369: ‘narrow’ should be ‘marrow’

Response: Corrected.

5. Line 372: change activate to generate.

Response: Done.

6. Line 383: ‘frist’ spelling error.

Response: Corrected.

7. Line 48: remove despite or reword sentence for grammar

Response: “Despite” was removed in the revised manuscript.

8. Line 50: current standard of care does not increase morbidity, if it did it would not be the standard. Chemotherapy and radiation can certainly decrease quality of life however.

Response: We revised the statement as “they tend to decrease the life quality of patients” in Page 3 Line 48-50 of the revised manuscript.

REVIEWERS' COMMENTS:

Reviewer #1 (Remarks to the Author):

The authors have well addressed my comments.

Reviewer #2 (Remarks to the Author):

The authors have adequately addressed this reviewer's comments.

Response to the Reviewers

Reviewer #1 (Remarks to the Author):

The authors have well addressed my comments.

Reviewer #2 (Remarks to the Author):

The authors have adequately addressed this reviewer's comments.

Response: We appreciate all the efforts of both the reviewers helping us to improve the manuscript.